# Learning State Representations from Random Deep Action-conditional Predictions

**Zeyu Zheng**
University of Michigan
`zeyu@umich.edu`

**Vivek Veeriah**
University of Michigan
`vveeriah@umich.edu`

**Risto Vuorio**
University of Oxford
`risto.vuorio@cs.ox.ac.uk`

**Richard Lewis**
University of Michigan
`rickl@umich.edu`

**Satinder Singh**
University of Michigan
`baveja@umich.edu`

## Abstract

Our main contribution in this work is an empirical finding that random General Value Functions (GVFs), i.e., deep action-conditional predictions—random both in what feature of observations they predict as well as in the sequence of actions the predictions are conditioned upon—form good auxiliary tasks for reinforcement learning (RL) problems. In particular, we show that random deep action-conditional predictions when used as auxiliary tasks yield state representations that produce control performance competitive with state-of-the-art hand-crafted auxiliary tasks like value prediction, pixel control, and CURL in both Atari and DeepMind Lab tasks. In another set of experiments we stop the gradients from the RL part of the network to the state representation learning part of the network and show, perhaps surprisingly, that the auxiliary tasks alone are sufficient to learn state representations good enough to outperform an end-to-end trained actor-critic baseline. We opensourced our code at `https://github.com/Hwhitetooth/random_gvfs`.

## 1 Introduction

Providing auxiliary tasks to Deep Reinforcement Learning (Deep RL) agents has become an important class of methods for driving the learning of state representations that accelerate learning on a main task. Existing auxiliary tasks have the property that their *semantics* are fixed and carefully designed by the agent designer. Some notable examples include pixel control, reward prediction, termination prediction, and multi-horizon value prediction (these are reviewed in more detail below). Unlike the prior approaches that require careful design of auxiliary task semantics, we explore here a different approach in which a set of *random action-conditional prediction tasks* are generated through a rich space of general value functions (GVFs) defined by a language of predictions of random features of observations conditioned on a random sequence of actions.

Our main, and perhaps surprising, contribution in this work is an empirical finding that auxiliary tasks of learning random GVFs—again, random in both predicted features and actions the predictions are conditioned upon— yield state representations that produce control performance that is competitive with state-of-the-art auxiliary tasks with hand-crafted semantics. We demonstrate this competitiveness in Atari games and DeepMind Lab tasks, comparing to multi-horizon value prediction [8], pixel control [13], and CURL [15] as our baseline auxiliary tasks. Note that while we present a reasonable approach to generating the semantics of the random GVFs we employ in our experiments, the specifics of our approach is not by itself a contribution (and thus not evaluated against other approaches to producing semantics for random GVFs), and alternative reasonable approaches for generating random GVFs could do as well.

35th Conference on Neural Information Processing Systems (NeurIPS 2021).

Additionally, through empirical analyses on illustrative domains we show the benefits of exploiting the richness of GVFs—their temporal depth and action-conditionality. We also provide direct evidence that using random GVFs learns useful representations for the main task through *stop-gradient* experiments in which the state representations are trained *solely* via the random-GVF auxiliary tasks without using the usual RL learning with rewards to influence representation learning. We show that, again, surprisingly, these stop-gradient agents outperform the end-to-end-trained actor-critic baseline.

## 2 Background and Related Work

**Horde and PSRs.** Auxiliary tasks were formalized and introduced to RL in [27] through the Horde architecture. Horde is an off-policy learning framework for learning knowledge represented as GVFs from an agent's experience. Our work is related to Horde in the use a rich subspace of GVF predictions but differs in that our interest is in the effect of learning these auxiliary predictions on the main task via shared state representations rather than to show the knowledge captured in these GVFs. Our work is also related to predictive state representations (PSRs) [17, 24]. PSRs use predictions *as* state representations whereas our work learns latent state representations from predictions. Recently, in the use of deep neural networks in RL as powerful function approximators, various auxiliary tasks have been proposed to improve the latent state representations of Deep RL agents. We review these auxiliary tasks below. Our work belongs to this family of work in that the auxiliary prediction tasks are used to improve the state representations of Deep RL agents.

**Auxiliary tasks using predefined GVF targets.** *UNREAL* [13] uses reward prediction and pixel control and achieved a significant performance improvement in DeepMind Lab but only marginal improvement in Atari. Termination prediction [14] is shown to be an useful auxiliary task in episodic RL settings. SimCore DRAW [9] learns a generative model of future observations conditioned on action sequences and uses it as an auxiliary task to shape the agent's belief states in partially observable environments. Fedus et al. [8] found that simply predicting the returns with multiple different discount factors (MHVP) serves as effective auxiliary tasks. MHVP relies on the availability of rewards and thus is different from our work and other unsupervised auxiliary tasks.

**Information-theoretic auxiliary tasks.** Information-theoretic approaches to auxiliary tasks learn representations that are informative about the future trajectory of these representations as the agent interacts with the environment. CPC [30], CPC|action [10], ST-DIM [1], DRIML [19], and ATC [25] apply different forms of temporal contrastive losses to learn predictions in a latent space. CURL [15] ignores the long-term future and applies a contrastive loss on the stack of consecutive frames to learn good visual representations. PBL [11] focuses on partially observable environments and introduces a separate target encoder to set the prediction targets. The target encoder is trained to distill the learned state representations. SPR [23] replaces the target encoder in PBL with a moving average of the state representation function. In addition to being predictive, PI-SAC [16] also enforces the state representations to be compressed. SPR and PI-SAC focuses on data efficiency and only conducted experiments under low data budgets. In these information-theoretic approaches, the targets are not GVFs and despite some empirical success, none could learn long-term predictions effectively. This is in contrast to GVF-like predictions which can be effectively learned via TD as in our work as well as the work presented above.

**Theory.** A few recent works [3, 7, 18] have studied the optimal representation in RL from a geometric perspective and provided theoretical insights into why predicting GVF-like targets is helpful in learning state representations. Our work is consistent with this theoretical motivation.

**GVF discovery.** Veeriah et al. [31] used metagradients to discover simple GVFs (discounted sums of features of observations) In this work, we show that random choices of features and random but rich GVFs are competitive with the state-of-the art of hand-crafted GVFs as auxiliary tasks.

**GVF RNNs.** Rather than using GVFs as auxiliary tasks, General Value Function Networks (GVFNs) [22] are a new family of recurrent neural networks (RNNs) where each dimension of the hidden state is a GVF prediction. GVFNs are trained by TD instead of truncated backprop through time. Our work relates to GVFNs in that both works use GVFs to shape state representations. However, unlike GVFNs, our work uses GVFs as auxiliary tasks and does not enforce any semantics to the state representations. Moreover, our empirical study mainly focuses on the control setting where the agent needs to maximize its long-term cumulative rewards, whereas [22] mainly focuses on time series modelling tasks and online prediction tasks and demonstrated the superior performance of GVFNs over conventional RNNs when the truncated input sequences are short during training.

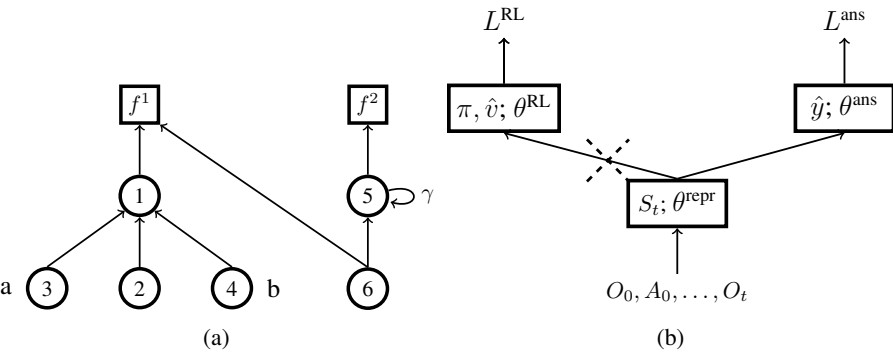

Figure 1: **(a)** An example of a question network. The squares represent *feature nodes* and circles represent *prediction nodes*. **(b)** The agent architecture. The dashed cross denotes an optional `stop-gradient` operation.

## 3 Method

In this section we first describe the specific GVFs we studied in this work. Then we describe an algorithm for the construction of random GVFs. We finish this section with a description of the agent architecture used in our empirical work.

### 3.1 GVFs with Interdependent TD Relationships

In this work, we study auxiliary prediction tasks where the semantics of the predictions are defined by a set of GVFs with interdependent TD relationships (this family of GVFs are often referred as *temporal-difference networks* in the literature [28, 29]). The TD relationships among the GVFs can be described by a graph with directional edges, which we call the *question network* as it defines the semantics of the predictions.

Figure 1a shows an example of a question network. The two squares represent two *feature nodes* and the six circles represent six *prediction nodes*. Node 1 (labeled in the circles) predicts the expected value of feature $f^1$ at the next step. Implicitly, this prediction is conditioned on following the current policy. Node 2 predicts the expected value of node 1 at the next step. Note that we can "unroll" the target of node 2 to ground it on the features. In this example, node 2 predicts the expected value of feature $f^1$ after two steps when following the current policy. Node 5 has a self-loop and predicts the expectation of the discounted sum of feature $f^2$ with a discount factor $\gamma$. We call node 5 a *discounted sum* prediction node. Node 3 is labeled by action $a$. It predicts the expected value of node 1 at the next step given action $a$ is taken at the current step. We say node 3 is *conditioned* on action $a$. Similarly, node 4 predicts the same target but is conditioned on action $b$. Node 6 has two outgoing edges. It predicts the *sum* (in general a weighted sum, but in this paper we do not explore the role of these weights and instead fix them to be 1) of feature $f^1$ and the value of node 5, both at the next step. In this case, it is hard to describe the semantic of node 6's prediction in terms of the features, but we can see that the prediction is still grounded on feature $f^1$ and $f^2$.

Generalising from the example above, a question network with $n_p$ prediction nodes and $n_f$ feature nodes defines $n_p$ predictions of $n_f$ features. We use $N_p$ to denote the set of all prediction nodes and $N_f$ to denote the set of all feature nodes. Let $W$ be the adjacency matrix of the question network. $W_{ij}$ denotes the weight on the edge from node $i$ to node $j$. We define $W_{ij} \triangleq 0$ if there is no edge from node $i$ to node $j$. Now consider an agent interacting with the environment. At each step $t$, it receives an observation $O_t$ and takes an action $A_t$ according to its policy $\pi$. Then at the next step it receives an observation $O_{t+1}$. The feature $f^k(O_t, A_t, O_{t+1})$ is a scalar function of the transition. The agent makes a prediction $\hat{y}^i(O_0, A_0, \ldots, O_t)$ for each prediction node $i$ based on its history; this is computed by a neural network in our work. For brevity, we use $f^k_{t+1}$ and $\hat{y}^i_t$ to denote $f^k(O_t, A_t, O_{t+1})$ and $\hat{y}^i(O_0, A_0, \ldots, O_t)$ respectively. The target for prediction $i$ at step $t$ is

denoted by $y_t^i$. If prediction node $i$ is not conditioned on any action, its target is

$$y_t^i = \mathbb{E}_\pi \Big[ \sum_{j \in N_p} W_{ij} y_{t+1}^j + \sum_{k \in N_f} W_{ik} f_{t+1}^k \Big]$$

otherwise, if it is conditioned on action $a^i$, its target is

$$y_t^i = \mathbb{E}_\pi \Big[ \sum_{j \in N_p} W_{ij} y_{t+1}^j + \sum_{k \in N_f} W_{ik} f_{t+1}^k | A_t = a^i \Big].$$

By the construction of the targets, the agent can learn the prediction $\hat{y}_t^i$ via TD. If $i$ is not conditioned on any action, then $\hat{y}_t^i$ is updated by

$$\hat{y}_t^i \leftarrow \sum_{j \in N_p} W_{ij} \hat{y}_{t+1}^j + \sum_{k \in N_f} W_{ik} f_{t+1}^k$$

otherwise, if $i$ is conditioned on action $a^i$, then $\hat{y}_t^i$ is updated by

$$\hat{y}_t^i \leftarrow \begin{cases} \sum\limits_{j \in N_p} W_{ij} \hat{y}_{t+1}^j + \sum\limits_{k \in N_f} W_{ik} f_{t+1}^k & \text{if } A_t = a^i \\ \hat{y}_t^i & \text{otherwise} \end{cases}$$

In an episodic setting, if the episode terminates at step $T$, we define $y_T^i \triangleq 0$ and $\hat{y}_T^i \triangleq 0$ for all prediction nodes $i$.

These GVFs represent a broad class of predictions. Many existing auxiliary prediction tasks can be expressed by a question network. Reward prediction [13] can be represented by a question network with a single feature node representing the reward and a single prediction node predicting the reward. Multi-horizon value prediction [8] can be represented by a similar question network but with multiple self-loop prediction nodes with different discounts. Termination prediction [14] can be represented by a question network with a feature node of constant 1 and a self-loop node with discount 1.

### 3.2 A Random Question Network Generator

In this work, instead of hand-crafting a new question network instance as in previous work on the use of predictions for auxiliary tasks, we verify a conjecture that a large number of random deep, action-conditional predictions is enough to drive the learning of good state representations without needing to carefully hand-design the semantics of those predictions. To test this conjecture, we designed a generator of random question networks from which we can take samples and evaluate their performance as auxiliary tasks. Specifically, we designed a heuristic algorithm that generates question networks with random features and random structures.

**Random Features.** We use random features, each computed by a scalar function $g^k$ with random parameters. For any transition $(O_t, A_t, O_{t+1})$, the feature is computed as $f_{t+1}^k = |g^k(O_{t+1}) - g^k(O_t)|$. Instead of directly using the output of $g^k$ as the feature, we use the amount of change in $g^k$. A similar transformation was used in pixel control [13].

**Random Structure.** The intuition behind our heuristic for generating random question network structures is to create predictions that correspond to executing an open-loop action sequence and then following the policy while accumulating one random feature at each time step. As we will illustrate in Section 4, these predictions can provide rich training signals for learning good representations. Specifically, the generator takes 5 arguments as input: number of features $n_f$, the discrete action set $\mathcal{A}$, a discount factor $\gamma$, depth $D$, and repeat $R$. Its output is a question network that contains $n_f$ feature nodes as defined above, and $D + 1$ layers, each layer contains $R \times |\mathcal{A}|$ prediction nodes except the first layer which contains $n_f$ prediction nodes. We construct the question network layer by layer incrementally from layer 0 to layer $D$. First, layer 0 has $n_f$ feature nodes and $n_f$ prediction nodes; each prediction node has an edge to a distinct feature node with weight 1 on the edge and has a self-loop with weight $\gamma$. Each prediction node in layer 0 predicts the discounted sum of its corresponding feature and are not conditioned on actions. Then for each layer $l$ ($1 \le l \le D$), we create $R \times |\mathcal{A}|$ prediction nodes. Each prediction node is conditioned on one action and there are exactly $R$ nodes that are conditioned on the same action. Each prediction node has two edges, one to a random node in layer $l - 1$ and one to a random feature node in layer 0. Note that prediction

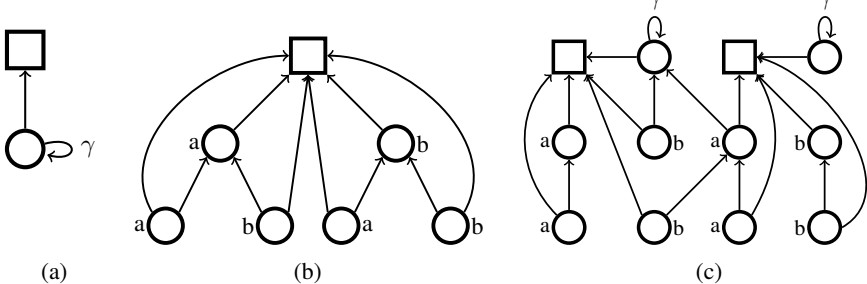

Figure 2: The question networks we studied in our illustrative experiment. **(a)** A discounted sum prediction. **(b)** A depth-2 tree question network with 2 actions. The bottom right prediction node predicts the sum of the values of the feature in the next two steps if action $b$ were taken for both the current and the next step. Other prediction nodes have similar semantics. **(c)** A random question network sampled from rGVFs. There are 2 features and 2 actions, with depth and repeat set to 2.

nodes in layer $1$ do not necessarily connect to a self-loop prediction node in layer $0$ - they may only connect to a feature node. A constraint for preventing duplicated predictions is included so that any two prediction nodes in layer $l$ that are conditioned on the same action cannot connect to the same prediction node in layer $l - 1$. In our preliminary experiments, we tried adding self-loops to deeper layers and allowing denser connections between nodes. Sometimes those additional loops and dense edges caused instability during training. We leave the study of more sophisticated question network structures to future work. The Appendix includes pseudocode for the random generator algorithm.

### 3.3 Agent Architecture

We used a standard auxiliary-task-augmented agent architecture, as shown in Figure 1b. We base our agent on the actor-critic architecture and it consists of 3 modules. The state representation module, parameterized by $\theta^{\text{repr}}$, maps the history of observations and actions $(O_0, A_0, \ldots, O_t)$ to a state vector $S_t$. The RL module, parameterized by $\theta^{\text{RL}}$, maps the state vector $S_t$ to a policy distribution over the available actions $\pi(\cdot|S_t)$ and an approximated value function $\hat{v}(S_t)$. The answer network module, parameterized by $\theta^{\text{ans}}$, maps the state vector $S_t$ to a set of predictions $\hat{y}(S_t)$ equal in size to the number of prediction nodes in the question network. Like in previous work, the augmented agent has more parameters than the base A2C agent due to the answer network module. However, the policy space and the value function space remain the same and these auxiliary parameters are only used for providing richer training signals for the state representation module.

We trained the network in two separate ways. In the auxiliary task setting, the RL loss $\mathcal{L}^{\text{RL}}$ is backpropagated to update the parameters of the state representation ($\theta^{\text{repr}}$) and the RL ($\theta^{\text{RL}}$) modules, while the answer network loss $\mathcal{L}^{\text{ans}}$ is backpropagated to update the parameters of the answer network ($\theta^{\text{ans}}$) and state representation ($\theta^{\text{repr}}$) modules. Note that the answer network loss only affects the RL module indirectly through the shared state representation module. In the stop-gradient setting, we stopped the gradients from the RL loss from flowing from $\mathcal{L}^{\text{RL}}$ to $\theta^{\text{repr}}$. This allows us to do a harsher and more direct evaluation of how well the auxiliary tasks can train the state representation without any help from the main task. For $\mathcal{L}^{\text{RL}}$, we used the standard actor-critic objective with an entropy regularizer. For $\mathcal{L}^{\text{ans}}$, we used the mean-squared loss for all the targets and predictions.

## 4 Illustrating the Benefits of Deep Action-conditional Questions

The main aim of the experiments in this section is to illustrate how deep action-conditional predictions can yield good state representations. We first use a simple grid world to visualize the impact of depth and action conditionality on the learned state representations. Then we demonstrate the practical benefit of exploiting both of these two factors by an ablation study on six Atari games. In addition, to test the robustness of the control performance to the hyperparameters of the random GVFs, we conducted a random search experiment on the Atari game Breakout.

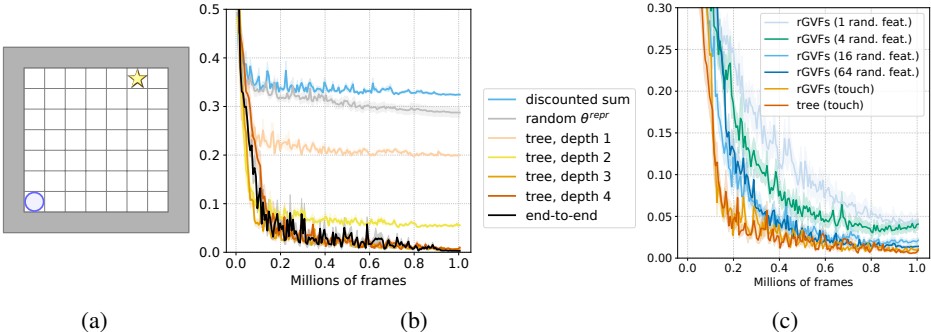

(a)           (b)           (c)

Figure 3: **(a)** The illustrative grid world environment. The blue circle denotes the agent and the yellow star denotes the rewarding state. **(b)** MSE between the learned value function and the true value function in the tree question networks experiment. **(c)** MSE between the learned value function and the true value function in the random question networks experiment.

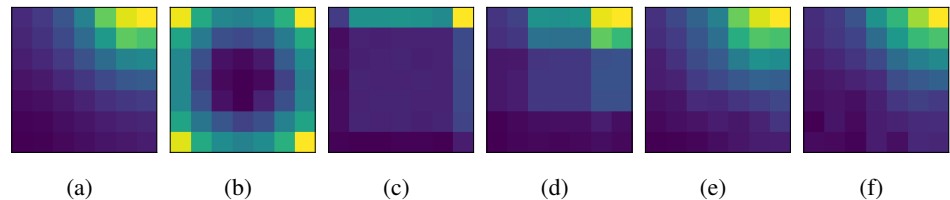

(a)      (b)      (c)      (d)      (e)      (f)

Figure 4: Visualization of the learned value functions in the empty room environment. Bright indicates high value and dark indicates low value. **(a)** The true values. **(b)** The discounted sum predictions of the `touch` feature. **(c)** - **(f)** The prediction are defined by a full-tree-structured question network regarding the `touch` feature. The depth of the tree increases from 1 to 4 from **(c)** to **(f)**.

## 4.1 Benefits of Depth and Action-conditionality: Illustrative Grid World

Although our primary interest (and the focus of subsequent experiments) is learning good policies, in this domain we study *policy evaluation* because this simpler objective is sufficient to illustrate our points and we can compute and visualize the true value function for comparison. Figure 3a shows the environment, a 7 by 7 grid surrounded by walls. The observation is a top-down view including the walls. There are 4 available actions that move the agent horizontally or vertically to an adjacent cell. The agent gets a reward of 1 upon arriving at the goal cell located in the top row, and 0 otherwise. This is a continuing environment so achieving the goal does not terminate the interaction. The objective is to learn the state-value function of a *random policy* which selects each action with equal probability. We used a discount factor of $0.98$.

Specifying a question network requires specifying both the structure and the features. Later we explore random features, but here we use a single hand-crafted `touch` feature so that every prediction has a clear semantics. `touch` is 1 if the agent's move is blocked by the wall and is 0 otherwise.

Using the `touch` feature we constructed two types of question networks. The first type is the discounted sum prediction of `touch` (we used a discount factor $0.8$) (Figure 2a). The second type is a *full action-conditional tree* of depth $D$. There is only one feature node in the tree which corresponds to the `touch` feature. Each internal node has 4 child nodes conditioned on distinct actions. Each prediction node also has a skip edge directly to the feature node (except for the child nodes of the feature node). Figure 2b shows an example of a depth-2 tree (the caption describes the semantics of some of the predictions). We also compared to a randomly initialized state representation module as a baseline where the state representation was fixed and only the value function weights were learned during training.

**Neural Network Architecture.** The empty room environment is fully observable and so the state representation module is a feed-forward neural network that maps the current observation $O_t$ to a state vector $S_t$. It is parameterized by a 3-layer multi-layer perceptron (MLP) with 64 units in the first two layers and 32 units in the third layer. The RL module has one hidden layer with 32

units and one output head representing the state value. (There is no policy head as the policy was given). The answer network module also has one hidden layer with 32 units and one output layer. We applied a stop-gradient between the state representation module and the RL module (Figure 1b). More implementation details are provided in the Appendix.

**Results.** We measured the performance by the mean-squared error (MSE) between the learned value function and the true value function across all states. The true value function was obtained by solving a system of linear equations [26]. Figure 3b shows the MSE during training. Both the random baseline and the discounted sum prediction target performed poorly. But even a tree question network of depth 1 (i.e., four prediction targets corresponding to the four action conditional predictions of `touch` after one step) performed much better than these two baselines. Performance increased monotonically with increasing depth until depth 3 when the MSE matched end-to-end training after 1 million frames.

Figure 4 shows the different value functions learned by agents with the different prediction tasks. Figure 4a visualizes the true values. Figure 4b shows the learned value function when the state representations are learned from discounted sum predictions of `touch`. Its symmetric pattern reflects the symmetry of the grid world and the random policy, but is inconsistent with the asymmetric true values. Figure 4c shows the learned value function when the state representations are learned from depth-1-tree predictions. It clearly distinguishes 4 corner states, 4 groups of states on the boundary, and states in the center area, as this grouping reflects the different prediction targets for these states.

For the answer network module to make accurate predictions of the targets of the question network, the state representation module must map states with the same prediction target to similar representations and states with different targets to different representations. As the question network tree becomes deeper, the agent learns finer state distinctions, until an MSE of 0 is achieved at depth 3 (Figure 4e).

## 4.2 Benefits of Random Question Nets: Illustrative Grid World

The previous experiment demonstrated benefits of temporally deeper action-conditonal prediction tasks. But achieving this by creating deeper and deeper full-branching action-conditional trees is not tractable as the number of prediction targets grows exponentially. The previous experiment also used a single feature formulated using domain knowledge; such feature selection is also not scalable. The random generator described in Section 3 provides a method to mitigate both concerns by growing random question networks with random features.

Specifically, we used *discount* 0.8, *depth* 4, and *repeat* equal to the number of features for generating random GVFs. Figure 3c shows the MSE of different random GVF variants. The performance of random GVFs with `touch`—that is, random but not necessarily full branching trees of depth 4—performed as well as `touch` with a full tree of depth 4. Random GVFs with a single random feature performed suboptimally; a random feature is likely less discriminative than `touch`. However, as the number of random features increases, the performance improves, and with 64 random features, random GVFs match the final performance of `touch` with a full depth 4 tree.

The results on the grid world provide preliminary evidence that random deep action-conditional GVFs with many random features can yield good state representations. We next test this conjecture on a set of Atari games, exploring again the benefits of depth and action conditionality.

## 4.3 Ablation Study of Benefits of Depth and Action Conditionality: Atari

Here we use six Atari games [4] (these six are often used for hyperparameter selection for the Atari benchmark [20]) to compare four different kinds of random GVF question networks: **(a)** random GVFs in which all predictions are *discounted sums* of distinct random features (illustrated in Figure 2a and denoted rGVFs-*discounted-sum* in Figure 5); **(b)** random GVFs in which all predictions are *shallow action-conditional* predictions, a set of depth-1 trees, each for a distinct random feature (denoted rGVFs-*shallow* in Figure 5); **(c)** random GVFs without action-conditioning (denoted rGVFs-*no-actions* in Figure 5); and **(d)** random GVFs that exploit both action conditionality and depth (illustrated in Figure 2c and denoted simply by rGVFs in Figure 5).

**Random Features for Atari.** The random function $g$ for computing the random features are designed as follows. The $84 \times 84$ observation $O_t$ is divided into 16 disjoint $21 \times 21$ patches, and a *shared* random linear function applies to each patch to obtain 16 random features $g_t^1, g_t^2, \ldots, g_t^{16}$. Finally, we process these features as described in §3.2.

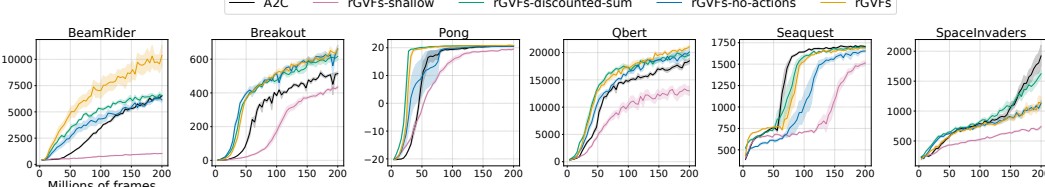

Figure 5: Learning curves of different question networks in six Atari games. x-axis denotes the number of frames and y-axis denotes the episode returns. Each curve is averaged over 5 independent runs with different random seeds. Shaded area shows the standard error.

**Neural Network Architecture.** We used A2C [20] with a standard neural network architecture for Atari [21] as our base agent. Specifically, the state representation module consists of 3 convolutional layers. The RL module has one hidden dense layer and two output heads for the policy and the value function respectively. The answer network has one hidden dense layer with 512 units followed by the output layer. We stopped the gradient from the RL module to the state representation module.

**Hyperparameters.** The discount factor, depth, and repeat were set to 0.95, 8, and 16 respectively. Thus there are $16 + 8 * 16 * |\mathcal{A}|$ total predictions. Random GVFs without action-conditioning has the same question network except that no prediction was conditioned on actions. To match the total number of predictions, we used $16 + 8 * 16 * |\mathcal{A}|$ random features for discounted sum and $8 * 16$ features for shallow action-conditional predictions. Additional random features were generated by applying more random linear functions to the image patches. The discount factor for discounted sum predictions is also 0.95. More implementation details are provided in the Appendix.

**Results.** Figure 5 shows the learning curves in the 6 Atari games. rGVFs-*shallow* performed the worst in all the games, providing further evidence for the value of making deep predictions. rGVFs consistently outperformed rGVFs-*no-actions*, providing evidence that action-conditioning is beneficial. And finally, rGVFs performed better than rGVFs-*discounted-sum* in 3 out of 6 games (large difference in BeamRider and small differences in Breakout and Qbert), was comparable in 2 the other 3 games, and performed worse in one—despite using many fewer features than rGVFs-*discounted-sum*. This suggests that structured deep action-conditional predictions can be more effective than simply making discounted sum predictions about many features.

### 4.4 Robustness and Stability

We tested the robustness of rGVFs with respect to its hyperparameters, namely discount, depth, repeat, and number of features. We explored different values for each hyperparameter independently while holding the other hyperparameters fixed to the values we used in the previous experiment. For each hyperparameter, we took 20 samples uniformly from a wide interval and evaluated rGVFs on Breakout using the sampled value. The results are presented in Figure 6. The lines of best fit (the red lines) in the left two panels indicate a positive correlation between the performance and the depth of the predictions, which is consistent with the previous experiments. Each hyperparameter has a range of values that achieves high performance, indicating that rGVFs are stable and robust to different hyperparameter choices. Additional results in BeamRider and SpaceInvaders are in the Appendix.

## 5 Comparison to Baseline Auxiliary Tasks on Atari and DeepMind Lab

In this section, we present the empirical results of comparing the performance of rGVFs against the A2C baseline [20] and three other auxiliary tasks, i.e., multi-horizon value prediction (MHVP) [8], pixel control (PC) [13], and CURL [15]. We conducted the evaluation in 49 Atari games [4] and 12 DeepMind Lab environments [2]. It is unclear how to apply CURL to partially observable environments which require long-term memory because CURL is specifically designed to use the stack of recent frames as the inputs. Thus we did not compare to CURL in the DeepMind Lab environments. Our implementation of rGVFs for this experiment is available at `https://github.com/Hwhitetooth/random_gvfs`.

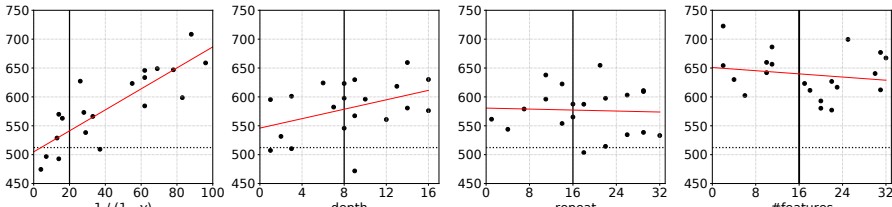

Figure 6: Scatter plots of scores in Breakout obtained by rGVFs with different hyperparameters. x-axis denotes the value of the hyperparameter. y-axis denotes the final game score after training for 200 million frames. The red line in each panel is the line of best fit. The dotted horizontal lines denote the performance of the end-to-end A2C baseline. The solid vertical lines denotes the values we used in our final experiments.

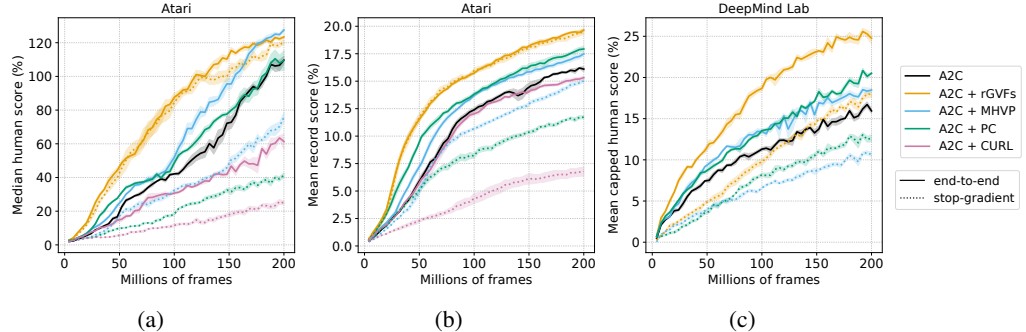

Figure 7: **(a)** Median human-normalized score across 49 Atari games. **(b)** Mean record-normalized score across 49 Atari games. **(c)** Mean capped human-normalized score across 12 DeepMind Lab environments. In all panels, the x-axis denotes the number of frames. Each dark curve is averaged over 5 independent runs with different random seeds. The shaded area shows the standard error.

**Atari Implementation.** We used the same architecture for rGVFs as in the prior section. For MHVP, we used 10 value predictions following [8]. Each prediction has a unique discount factor, chosen to be uniform in terms of their effective horizons from 1 to 100 ($\{0, 1 - \frac{1}{10}, 1 - \frac{1}{20}, \ldots, 1 - \frac{1}{90}\}$). The architecture for MHVP is the same as rGVFs. For PC, we followed the architecture design and hyperparameters in [13]. For CURL, we implemented it in our experiment setup by using the code accompanying the paper as a reference [1]. When not stopping gradient from the RL loss, we mixed the RL updates and the answer network updates by scaling the learning rate for the answer network with a coefficient $c$. We searched $c$ in $\{0.1, 0.2, 0.5, 1, 2\}$ on the 6 games in the previous section. $c = 1$ worked the best for all methods. More details are in the Appendix.

**DeepMind Lab Implementation.** We used the same RL module and answer network module as Atari but used a different state representation module to address the partial observability. Specifically, the convolutional layers in the state representation module were followed by a dense layer with 512 units and a GRU core [5, 6] with 512 units.

**Results.** Figure 7a and Figure 7b shows the results for both the stop-gradient and end-to-end architectures on Atari, comparing to two standard human-normalized score measures (median human-normalized score [21] and mean record-normalized score [12]). When training representations end-to-end through a combined main task and auxiliary task loss, the performance of rGVFs matches or substantially exceeds the three baselines. Although the original paper shows that CURL improves agent performance in the data-efficient regime (i.e., $100K$ interactions in Atari), our results indicate that it hurts the performance in the long run. We conjecture that CURL is held back by failing to capture long-term future in representation learning. Surprisingly, the stop-gradient rGVFs agents outperform the end-to-end A2C baseline, unlike stop-gradient versions of the baseline auxiliary task agents. Figure 7c shows the results for both stop-gradient and end-to-end architectures on 12 DeepMind Lab environments (using mean capped human-normalized scores). Again, rGVFs

---

[1]`https://github.com/aravindsrinivas/curl_rainbow`

substantially outperforms both auxiliary task baselines, and the stop-gradient version matches the final performance of the end-to-end A2C. Taken together the results from these 61 tasks provide substantial evidence that rGVFs drive the learning of good state representations, outperforming auxiliary tasks with fixed hand-crafted semantics.

## 6 Conclusion and Future Work

In this work we provided evidence that learning how to make random deep action-conditional predictions can drive the learning of good state representations. We explored a rich space of GVFs that can be learned efficiently with TD methods. Our empirical study on the Atari and DeepMind Lab benchmarks shows that learning state representations solely via auxiliary prediction tasks defined by random GVFs outperforms the end-to-end trained A2C baseline. Random GVFs also outperformed pixel control, multi-horizon value prediction, and CURL when being used as part of a combined loss function with the main RL task.

In this work, the question network was sampled before learning and was held fixed during learning. An interesting goal for future research is to find methods that can adapt the question network and discover useful questions during learning. The question networks we studied are limited to discrete actions. It is unclear how to condition a prediction on a continuous action. Thus another future direction to explore is to extend action-conditional predictions to continuous action spaces.

### Acknowledgement

This work was supported by DARPA's L2M program as well as a grant from the Open Philanthropy Project to the Center for Human Compatible AI. Any opinions, findings, conclusions, or recommendations expressed here are those of the authors and do not necessarily reflect the views of the sponsors.

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
