# OpenReview forum: "Learning State Representations from Random Deep Action-conditional Predictions"
_NeurIPS.cc/2021/Conference — NeurIPS 2021 Poster_

### Official Review · Reviewer_QPtM · 2021-07-06

**Rating:** 6
**Confidence:** 4

**Summary:**

This work proposes random General Value Functions (rGVFs) as auxiliary tasks to extract useful representations for reinforcement learning. The target feature and the connections between the prediction nodes are randomized. The proposed method outperforms end-to-end A2C and other baselines with auxiliary training in 49 Atari games and 12 DeepMind Lab environments.

**Limitations And Societal Impact:**

It is nice that the authors addressed the limitation of rGVFs to discrete actions. This limitation may be problematic in certain continuous control tasks. But referring to Figure 5, it seems that the performance of rGVFs-no-actions is on par with that of rGVFs except for BeamRider. It would be interesting to evaluate rGVFs-no-actions in continuous control tasks such as MuJoCo.

 Another limitation not mentioned by the authors is the computation load required. Since rGVFs perform auxiliary training with deep multiple graphs, it seems to bring a great increment in computation, especially for tasks with large action space (4 to18 for 49 Atari games). If I'm not mistaken the amount of computation is not described in the appendix, contrary to what is indicated in checklist 3(d).

The last limitation is the expressibility of the question network in complex state space. In Figure 7(a) and 7(b), the scores of rGVFs with and without stop-gradient are nearly the same. However, in DMLab experiment, rGVFs with stop-gradient are on par with end-to-end A2C. I think this gap is from the fact that DMLab is a POMDP, therefore it is more challenging to obtain rich state representation (from the GRU with the entire past trajectory) just using the random targets. There is a concern that the representation learned by rGVFs will not be effective in situations in more complex state space.

**Main Review:**


1.Originality

To my knowledge, it is the first paper in which both the target and the connections of the target network are randomized, so I think the originality is sufficient. The related work is adequately cited.

One missing related work that introduces compositional GVFs with action conditions and discounts.

[1] Andrew Patterson and Matthew Schlegel, A Comparison of General Value Functions and Temporal-Difference Networks, AAMAS 2016.


2. Quality

As this is a study without theoretical results, the quality in terms of the experimental results was mainly reviewed. I have a very mixed feeling about the experimental results. The empirical gain in Figure 7 is impressive, but the reasons for such improvement are not well discussed.

2.1) Significant empirical gain

The performance gain of rGVFs over the end-to-end A2C in Figure 7 is impressive. Especially it is a very valuable result that the stop-gradient version of rGVFs outperforms the Vanilla A2C in Atari. This is a meaningful result because the experiment was conducted on a large number of tasks (49 in Atari and 12 in DMLab) without cherry-picking specific tasks that are advantageous to their method.

A2C is a fairly weak baseline and its performance is far below the current state of the art (e.g., MuZero and R2D2). However, since rGVFs are orthogonal to the RL training, it is expected to bring similar performance gain when combined with SOTA methods. The strength of this paper will be emphasized even more by adding such empirical results.

2.2) Lack of intuitive explanation and empirical analysis.

The greatest weakness of this work is that it only lists the empirical gains and lacks a detailed analysis of why such gain is achieved.

The illustrative result in Figure 3b is appreciated. It makes sense that the random $\theta^{repr}$ suffers from high MSE since the value network has only one hidden layer and increasing the depth of the tree decreases the MSE as low as that of end-to-end. However, it does not provide intuition on why rGVFs should outperform end-to-end methods (as in Figure 7). This work will be largely strengthened with empirical analysis on what is learned to improve performance. Here are a few ideas that come to mind:

- Analysis of successful random graphs: since the graphs are randomly generated, It seems that certain types of random graphs may be more effective than others (maybe graphs similar to TD are helpful? or semantics as 'touch' in Figure 2?). Maybe the improvement is from the restricted structures of the graph. Although the authors name their work as 'random' GVF, it seems that the structure is largely restricted (2 connections for each prediction nodes, self-loop in layer 0, each layer containing the same number of nodes based on R and D, weighted sum with weight 1, ..etc.)

- Analysis of successful tasks: the effectiveness of rGVFs may vary depending on the properties of target tasks as the authors state in line 283. Also in Figure 2 in Appendix, it seems that the amount of gain varies from game to game.

I'm willing to raise my score if such analysis is provided.


3. Clarity

This work is generally well organized and the main claim of the work is well delivered. The source code in the supplementary material is superbly organized and provides enough information to reproduce the results. However, I think the overall clarity can be largely improved, and there seem to be some conflicts between the contents of the manuscript.

- Section 3.1 and 3.2 introduce the core ideas of this study but they are somewhat complicated to understand. Maybe providing a figure with all the notations ($f^k, y^i, n_f, n_p, z^{t}_{i}, W, R, D$) will make it much easier to understand the overall structure. Or simply providing example update equations for figure 1(a) with time index would be helpful.
- There are some sentences too long to understand (e.g. line 144). I'd appreciate it if the authors could break up such sentences.
- It seems that the input for the representation network is just $O_t$, not $O_0, A_0, \ldots, O_t$ if the environment is not POMDP. It would be great if there are corresponding explanations at the beginning of the paragraph in line 109 and in Figure 1b.
- Adding the vanilla A2C result in Figure 5 will be appreciated (plotting the results of Figure 2 in appendix)
- The star symbol in Figure 3a should be moved one cell to the right to match the results in Figure 4?
- Line 263 describes a depth-1 tree and refers Figure 2b, but it is confusing since the tree in Figure 2b seems to be a depth-2 tree without a self-loop.

**(Post rebuttal update)** I appreciate the detailed response and additional experimental results from the authors. It seems that the ablation study in the common response addresses my main concern on why rGVFs stop-gradient should outperform end-to-end A2C.  Therefore, I am willing to raise my rating from 5 to 6.

4. Significance

- The empirical result is significant and as the authors mention in the conclusion section, there is much room for improvement to generate more general value functions.
- Although the authors append rGVFs to A2C only, it seems to bring improvements when combined with other SOTA RL methods.
- The authors evaluate their work on Atari and DMLab environments, but I think rGVFs can be extended to solve generalization problems of RL since it may help to learn task-agnostic representations (e.g. pretrain rGVFs in source tasks and use the learned features to finetune different target tasks.)




**Time Spent Reviewing:**

20

---

> ### Author Response · Authors · 2021-08-10
> **Author Response to Reviewer QPtM**
>
> We thank the reviewer for the detailed comments. We address the specific comments here.
>
> > ...it does not provide intuition on why rGVFs should outperform end-to-end methods (as in Figure 7). This work will be largely strengthened with empirical analysis on what is learned to improve performance.
>
> rGVFs, like other auxiliary tasks, benefit RL agents by providing additional (denser) and richer learning signals than the sparse reward signal. We did a new ablation study which showed that dense training signals are important for good performance. Please see the common response for full details. Based on this insight, we conjecture that stop-gradient rGVFs outperformed end-to-end A2C in Atari because it received multiple training signals at every timestep whereas A2C only received sparse training signals from rewards; we expect that the state representations learned to answer the rGVF questions is also good for learning the optimal value function and policy.
>
> > [Clarity] ... Or simply providing example update equations for figure 1(a) with time index would be helpful.
>
> We will provide example update equations for Figure 1(a) in the revision. Thanks for the suggestion.
>
> > There are some sentences too long to understand (e.g. line 144). I'd appreciate it if the authors could break up such sentences.
>
> We will replace long sentences with concise ones in the revision. Specifically, we will replace Line 144 with "We designed the random question network generator based on the following intuition. Each prediction corresponds to executing an open-loop action sequence followed by the agent’s policy. Along the trajectory, it accumulates a feature-value (this would be the reward for the standard value function) at each timestep. Depending on the edges in the question network, the accumulated features can be different for different timesteps."
>
>
> > It seems that the input for the representation network is just $O_t$, not $O_0,A_0, …,O_t$ if the environment is not POMDP. It would be great if there are corresponding explanations at the beginning of the paragraph in line 109 and in Figure 1b.
>
> We will clarify this detail explicitly in the revision. Thanks!
>
> > Adding the vanilla A2C result in Figure 5 will be appreciated (plotting the results of Figure 2 in appendix)
>
> We will add a curve for A2C in Figure 5 in the revision. Thanks.
>
> > The star symbol in Figure 3a should be moved one cell to the right to match the results in Figure 4?
>
> The location of the star symbol in Figure 3(a) is correct. We broke the symmetry on purpose. Note that Figure 4(a) is also asymmetric (the two cells adjacent to the top-right corner have different colors).
>
> > Line 263 describes a depth-1 tree and refers Figure 2b, but it is confusing since the tree in Figure 2b seems to be a depth-2 tree without a self-loop.
>
> Figure 2(b) was meant to illustrate the tree structure rather than being a precise visualization of the question network we used in the experiments. We thought a depth-2 tree would serve better than a depth-1 tree as a representative example. We will clarify this confusion in Line 263 in the revision. Thanks!
>
> > Another limitation not mentioned by the authors is the computation load required...
>
> The computational overhead is minor, please see the common response.
>
> > The last limitation is the expressibility of the question network in complex state space. In Figure 7(a) and 7(b), the scores of rGVFs with and without stop-gradient are nearly the same. However, in DMLab experiment, rGVFs with stop-gradient are on par with end-to-end A2C. I think this gap is from the fact that DMLab is a POMDP, therefore it is more challenging to obtain rich state representation (from the GRU with the entire past trajectory) just using the random targets. There is a concern that the representation learned by rGVFs will not be effective in situations in more complex state space.
>
> Though we believe Atari and DMLab domains already have fairly complex state spaces, we agree it would be interesting in future work to explore more complex domains. We note however that although in DMLab rGVFs with stop-gradient are on par with end-to-end A2C, the stop-gradient experiments are an analysis tool;  in practice rGVFs would be used as auxiliary tasks in a combined task+auxiliary loss. In DMLab, A2C+rGVF significantly outperforms the baseline A2C with other auxiliary tasks, even though the stop-gradient performance did not dominate A2C alone.

---

### Official Review · Reviewer_7T1V · 2021-07-14

**Rating:** 7
**Confidence:** 4

**Summary:**

This paper introduces a new auxiliary task for learning state representations. This is done by predicting general value functions (GVFs), which are predictions of random features of observations conditioned on sequences of actions. The GVFs of interest can be represented with graphs which they call "question networks". The edges in this graph define nodes predicting the value of other nodes in a previous time-step, and many existing prediction tasks in the literature such as reward prediction, multi-horizon value prediction, and termination prediction can be represented using a question network.

The key contribution here is that predicting the value of nodes in a randomly generated question can provide a strong learning signal for learning representations. Several experiments are provided that show that: (1) representations learned by predicting the values of large random question networks (rGVF) outperform baselines like pixel-control and MHVP; (2) rGVF was the only method that learned good representations even when using a stop-gradient to prevent the RL learning signal from shaping the representation; and (3) several ablation tests were done to show the importance of action conditioning and depth.

**Ethical Concerns:**

I have no ethical concerns with this paper.

**Limitations And Societal Impact:**

The paper provides quite a generic analysis of the potential negative societal impact. I would like to see a bit more done here. For instance, while rGVFs can outperform hand designed auxiliary tasks, does this come at a cost in computational efficiency?

**Main Review:**

1. Originality: As this paper highlights, the idea of using GVFs as targets for auxiliary tasks is not new. However, for the most part the GVFs used in the past were hand defined. Work has been done on trying to discover useful GVFs, and this paper shows that interestingly, randomly generating large and deep enough GVFs can be enough for learning useful representations even in complex environments and tasks.
2. Quality: Overall, the quality of this paper is high. I understand that the experiments are mainly done to compare rGVFs with other GVFs. However, there has been significant recent work such as [1, 2, 3] that also shows that auxiliary tasks can accelerate learning or even separate representation learning (with stop-gradients) and still learn strong policies. The one major question that I have after reading this paper is: how does this method of representation learning compare to the SOTA approaches that use contrastive learning? This would be useful to know for practitioners looking to implement one of these algorithms.
3. Clarity: The clarity of the paper is generally high. I think one addition that would help readers understand the method would be an algorithm box which describes how the all the losses are generated from a batch of trajectories (my assumption is that the TD learning happens using the predictions from the answer network and that there are some stop-gradients in this process, but I want this to be explicitly stated).
I appreciate the clarity of the writing in the paper, and the experiment section stands out as being especially well-written.
4. Significance: Designing ways to improve representation learning in RL is an important problem that has received a lot of attention from researchers lately. This paper also gives an important result about the design of GVFs (randomly generating a large number of them can outperform a single hand-designed GVF).

[1] Srinivas, Aravind, et al. “CURL: Contrastive Unsupervised Representations for Reinforcement Learning.” ArXiv:2004.04136 [Cs, Stat], Sept. 2020. [arXiv.org](http://arxiv.org/), [http://arxiv.org/abs/2004.04136](http://arxiv.org/abs/2004.04136).

[2] Stooke, Adam, et al. “Decoupling Representation Learning from Reinforcement Learning.” ArXiv:2009.08319 [Cs, Stat], May 2021. [arXiv.org](http://arxiv.org/), [http://arxiv.org/abs/2009.08319](http://arxiv.org/abs/2009.08319).

[3] Yang, Mengjiao, and Ofir Nachum. “Representation Matters: Offline Pretraining for Sequential Decision Making.” ArXiv:2102.05815 [Cs], Feb. 2021. [arXiv.org](http://arxiv.org/), [http://arxiv.org/abs/2102.05815](http://arxiv.org/abs/2102.05815).

**Time Spent Reviewing:**

4

---

> ### Author Response · Authors · 2021-08-10
> **Author Response to Reviewer 7T1V**
>
> We thank the reviewer for the detailed comments. We address the specific comments here.
>
> > ...how does this method of representation learning compare to the SOTA approaches that use contrastive learning?
>
> We agree with the reviewer that such a comparison would be helpful and thank the reviewer for sharing the related works. However, none of the three recent works mentioned by the reviewer conducted experiments under the same setup as our work and so we cannot compare based on the published results in those papers -- thus we need to conduct extra experiments to make the comparison. We are actively working on adopting CURL to our setup and will provide a comparison between rGVF and CURL in the camera-ready version if accepted.
>
> > ...while rGVFs can outperform hand designed auxiliary tasks, does this come at a cost in computational efficiency?
>
> The computational overhead is minor, please see the common response.

---

> ### Author Response · Authors · 2021-09-01
> **Discussion**
>
> Thank you for your positive initial assessment! We would like to kindly remind you to read our rebuttal. If our rebuttal successfully addresses your concerns we hope that you would take that into account. Thanks!

---

> > ### Comment · Reviewer_7T1V · 2021-09-01
> > **Response**
> >
> > Thanks for the comments. I still believe this to be an interesting and strong submission and I think the score of 7 remains appropriate.

---

### Official Review · Reviewer_FnXv · 2021-07-18

**Rating:** 7
**Confidence:** 4

**Summary:**

Summary
-------

Owing to the importance of state representation in RL, this paper
provides an empirical investigation of random General Value Functions
(GVFs) for shaping state representation in RL. The paper briefly
outlines the random generation process for their GVFs before conducting
extensive experiments on GridWorld, as well as ablation studies in Atari
and DeepMind Lab environments. The authors find that sufficiently deep
random GVFs are able to provide state representations for policy
evaluation in a visual gridworld environment. Surprisingly, constructing
the state representation as an auxiliary task alone is enough to
outperform a baseline actor-critic on Atari and DeepMind lab.




**Limitations And Societal Impact:**

The authors adequately addressed the limitations and potential negative societal impact of their work.

**Main Review:**

Decision
--------

While the paper focuses on a relatively narrow problem regarding the
benefits of random GVFs as auxiliary tasks, the extensive investigation
and surprising findings deserves interest from the RL community. The
paper is held back by some clarity issues, and as such I am not able to
easily follow critical sections. Based on this, I am giving this paper a
5 (" Marginally below the acceptance threshold"). I have outlined below
some comments to further improve the paper. I am willing to raise my
score either after revisions or discussion with the authors / other
reviewers.

Strengths
---------

-   The contribution is well motivated and contextualized with respect
    to the current literature. State representation is an important
    problem in reinfo
rcement learning, and conventional methods using
    state representations that optimize a "deep RL objective" are known
    to produce representations that are not robust. With this in mind,
    it makes sense to investigate how good random features can be
    because intuition would leave us to think that random features are
    not good.
-   Extensive experiments covering simple environments (GridWorld),
    pixel control (Atari) and continuous control (Deepmind Lab). Because
    this is the main contribution of the paper, it is important that
    this is good. Both the breadth and depth of the experiments are
    commendable. Although I have some issues with specifics in the
    experiments, I think they are good overall

Weaknesses
----------

-   Explanations of general value functions and the random generation
    process is confusing or not given sufficient reasoning. I have
    outlined detailed comments below.
-   Some empirical findings are unclear. I have outlined specific
    concerns below as well.

Detailed Comments
-----------------

-   Line 121/123: I thought the targets $z_i$ would be used to update
    the predictions $y_i$, but they seem separate from $y_i$ and do not
    appear in the update. Moreover, you say that this is a TD update.
    The way that the update is written does not look like a TD update.
    Is it TD update because the update to $y^i_t$ has $y^i_{t+1}$ on the
    left hand side? This should be explained a bit more.
-   Line 115/140: The definition of feature here is a feature of the
    transition and not the observation. This seems like an important
    characterization and should merit some discussion. You point to
    Jaderberg et al. (2016) but I am unable to find any justification in
    that paper either.
-   Lines 144-165: While I am able to understand the generation process
    after a couple reads of this paragraph, I think there is a better
    way to convey such information. The pseudocode in the appendix is a
    bit more readable and, if possible, should be condensed using the
    plain english of the paragraph.
-   Line 152: Is it necessary to have exactly one prediction node per
    feature node, or is this a design choice?
-   Line 156: Why are R repeats necessary for each action prediction
    node?
-   Line 157: Why must the prediction nodes at layer l need to be
    connected to both a prediction node in layer 0 and layer l - 1? I
    don't see why these skip connections are critical for the generation
    process.
-   Line 200: Why is the gridworld environment treated as a continuing
    environment and not episodic? Is the learning curve in Figure 3b/c
    from one single stream of experience?
-   Figure 6: Dont the scatter plots for depth/repeat/num.features
    contradict the findings in the gridworld experiments? The scatter
    plots suggest that there is no trend in performance and these
    hyperparameters? Also, why is there such a large difference in
    performance between similar scatter points (e.g. the two left most
    points in the depth plot)?

Minor Comments
--------------

-   Figure 1a: I think the action label for nodes 3 and 4 should be in
    the circle to denote that they are part of that node. Without the
    text explaining the figure, it makes information hard to read off. I
    would suggest that nodes 3 and 4 be relabelled to 3 \| a and 4 \| b.
-   It is unclear whether the visual gridworld is a good environment
    choice for evaluating GVFs. For one, the feature set is not entirely
    meaningful as the states / observations provided to the agent is the
    entire grid as opposed to some local region of the agent. Second,
    while the environment seems simple, the number of iterations needed
    to learn anything meaningful is on the order of hundreds of
    thousands.

Post Rebuttal
--------------
In light of the discussion below, I have increased my score from a 5 -> 7.

**Time Spent Reviewing:**

8

---

> ### Author Response · Authors · 2021-08-10
> **Author Response to Reviewer FnXv**
>
> We thank the reviewer for the detailed comments. We address the specific comments here.
>
> > Line 121/123: I thought the targets $z^i$ would be used to update the predictions $y^i$, but they seem separate from $y^i$ and do not appear in the update. Moreover, you say that this is a TD update. The way that the update is written does not look like a TD update. Is it TD update because the update to $y_t^i$ has $y_{t+1}^i$ on the left hand side? This should be explained a bit more.
>
> $z^i$ defines the true value of the prediction. However, in RL we do not observe the true value directly but can only estimate it instead. Line 121/123 describes how the TD estimate is constructed. It is TD because it “learns a guess from a later guess”. In contrast to conventional TD, the TD target for question $i$ at time $t$ builds on the estimated answer of all questions at time $t+1$ rather than only building on the estimated answer of the same question at time $t+1$. This more factored form of TD is well discussed in [Sutton et al, 2004].
>
> > Line 115/140: The definition of feature here is a feature of the transition and not the observation. This seems like an important characterization and should merit some discussion. You point to Jaderberg et al. (2016) but I am unable to find any justification in that paper either.
>
> We define a feature as a function of the transition rather than just the observation to be more general and thus cover a broader range of features. Regarding the feature we chose in Line 140, which followed Jaderberg et al, [2016], one intuition behind this specific design choice is that taking the difference could potentially highlight the change in consecutive observations and ignore the irrelevant static background. We believe this makes the features more informative.
>
> > Lines 144-165: While I am able to understand the generation process after a couple reads of this paragraph, I think there is a better way to convey such information. The pseudocode in the appendix is a bit more readable and, if possible, should be condensed using the plain english of the paragraph.
>
> We like the idea of condensing the pseudocode with English and presenting it in the main text. We will implement this in a revision. Thanks.
>
> > Line 152: Is it necessary to have exactly one prediction node per feature node, or is this a design choice?
>
> It is a design choice. It is possible to have multiple prediction nodes per feature, each with a different discount factor like multi-horizon value prediction. We chose to keep it simple by only allowing 1 prediction node per feature.
>
> > Line 156: Why are R repeats necessary for each action prediction node?
>
> It is a design choice we made to allow more flexibility. Note that these $R$ nodes may connect to different feature nodes and different nodes in the previous layer though they are all conditioned on the same action, thus they have different semantics. We thought it might be limiting to only allow one node per action at each depth.
>
> > Line 157: Why must the prediction nodes at layer $l$ need to be connected to both a prediction node in layer 0 and layer $l - 1$? I don't see why these skip connections are critical for the generation process.
>
> The skip connections provide dense training signals at every timestep. We did a new ablation study to confirm the importance of skip connections. Please see the common response for more details including results of a new experiment.
>
> > Line 200: Why is the gridworld environment treated as a continuing environment and not episodic? Is the learning curve in Figure 3b/c from one single stream of experience?
>
> There is no specific reason why it must be continuing or episodic. We believe that the results would hold in an episodic setting as well. Each curve in Figure 3b/c is an average over 10 independent runs. Each run employed 8 parallel actors to generate the data.
>
> > Figure 6: Don't the scatter plots for depth/repeat/num.features contradict the findings in the gridworld experiments? The scatter plots suggest that there is no trend in performance and these hyperparameters? Also, why is there such a large difference in performance between similar scatter points (e.g. the two left most points in the depth plot)?
>
> In hindsight, Breakout was not an ideal choice for testing the robustness of the algorithm. As shown in Figure 6, Breakout shows little variance in performance unless the auxiliary predictions are extremely shallow (leftmost plot using discount). We repeated the robustness experiment on BeamRider and SpaceInvaders. For each dimension (i.e., discount, depth, repeat, and number of features), we randomly sampled 10 values while keeping the other dimensions to their default values. We then drew the same scatter plot as Figure 6 and did linear regression with the hyperparameter as the variables and the performance as the target. The coefficients of the linear regression are listed in the table below.
>
> |       | $1 / (1 - \gamma)$ | depth | repeat | number of features |
> |:-:|:-:|:-:|:-:|:-:|
> | Breakout | 52.2 | 19.0 | -1.8 | -6.6 |
> | BeamRider | 510.8 | 871.5 | 147.6 | 581.0 |
> | SpaceInvaders | 28.1 | 191.0 | 183.2 | 17.7 |
>
>
> The positive coefficients indicate deeper predictions and more random features yield better performance, which is consistent with the gridworld experiment (and as seen in the table above BeamRider and SpaceInvaders make this case much better than Breakout). Moreover, as shown by the raw scores in the tables below, there is a wide range of values for each hyperparameter that yields good performance, demonstrating the robustness of the rGVFs to the hyperparameters of the random generator.
>
> | $1 / (1 - \gamma)$ | 3 | 9 | 30 | 35 | 48 | 62 | 64 | 91 | 93 | 93 | A2C |
> | :-: | :-: | :-: | :-: | :-: | :-: | :-: | :-: | :-: | :-: | :-: | :-: |
> | BeamRider | 9500.96 | 9111.7 | 10097.62 | 10944.22 | 11875.14 | 8054.6 | 11692.02 | 11979.5 | 9707.62 | 11546.68 | 6534.492 |
>
> | depth | 2 | 3 | 4 | 5 | 8 | 10 | 10 | 11 | 15 | 16 | A2C |
> | :-: | :-: | :-: | :-: | :-: | :-: | :-: | :-: | :-: | :-: | :-: | :-: |
> | BeamRider | 7840.38 | 8531.72 | 9518.26 | 9263.7 | 9573.3 | 13122.4 | 8965.22 | 6824.72 | 10973.6 | 11618.4 | 6534.492 |
>
> | repeat | 1 | 2 | 4 | 4 | 11 | 14 | 21 | 28 | 29 | 30 | A2C |
> | :-: | :-: | :-: | :-: | :-: | :-: | :-: | :-: | :-: | :-: | :-: | :-: |
> | BeamRider | 9536.8 | 10452.24 | 7968.34 | 11007.86 | 11527.58 | 9299.94 | 8164.32 | 10265.68 | 11752.9 | 9555.24 | 6534.492 |
>
> | number of features | 3 | 4 | 7 | 8 | 10 | 10 | 21 | 26 | 28 | 29 | A2C |
> | :-: | :-: | :-: | :-: | :-: | :-: | :-: | :-: | :-: | :-: | :-: | :-: |
> | BeamRider | 10670.7 | 10984.64 | 12655.68 | 10450.88 | 11122.3 | 11098.9 | 13239.92 | 12792.38 | 11592.86 | 12379.28 | 6534.492 |
>
> | $1 / (1 - \gamma)$ | 14 | 14 | 38 | 40 | 44 | 46 | 64 | 68 | 84 | 90 | A2C |
> | :-: | :-: | :-: | :-: | :-: | :-: | :-: | :-: | :-: | :-: | :-: | :-: |
> | SpaceInvaders | 901.15 | 806.05 | 1387.55 | 1662.7 | 881.35 | 1511.05 | 1284.85 | 952.4 | 931.7 | 1213.3 | 1917.960 |
>
> | depth | 1 | 1 | 3 | 6 | 6 | 7 | 8 | 11 | 15 | 15 | A2C |
> | :-: | :-: | :-: | :-: | :-: | :-: | :-: | :-: | :-: | :-: | :-: | :-: |
> | SpaceInvaders | 832.9 | 767.45 | 866.95 | 990.75 | 924.6 | 899.2 | 832.6 | 1096.75 | 1292.9 | 1456.25 | 1917.960 |
>
> | repeat | 2 | 3 | 4 | 7 | 10 | 15 | 22 | 22 | 22 | 31 | A2C |
> | :-: | :-: | :-: | :-: | :-: | :-: | :-: | :-: | :-: | :-: | :-: | :-: |
> | SpaceInvaders | 776.7 | 624.05 | 800.6 | 876.9 | 784.0 | 999.8 | 1171.0 | 951.0 | 1162.05 | 1282.35 | 1917.960 |
>
> | number of features | 3 | 4 | 7 | 13 | 18 | 21 | 21 | 25 | 26 | 27 | A2C |
> | :-: | :-: | :-: | :-: | :-: | :-: | :-: | :-: | :-: | :-: | :-: | :-: |
> | SpaceInvaders | 786.9 | 947.0 | 834.9 | 905.9 | 848.2 | 797.55 | 850.1 | 928.65 | 1036.15 | 826.75 | 1917.960 |
>
> Regarding the variance in the scatter plot, it is known that the performance of deep RL agents in Atari has high variance due to stochastic exploration, stochastic initial state, etc. It is hard to draw any solid conclusions from each individual data point in the plot but the overall trend captured by the coefficients of the linear regression fits is informative.
>
> > Figure 1a: I think the action label for nodes 3 and 4 should be in the circle to denote that they are part of that node. Without the text explaining the figure, it makes information hard to read off. I would suggest that nodes 3 and 4 be relabelled to 3 | a and 4 | b.
>
> We will adjust the diagram accordingly in the revision. Thanks!
>
> > It is unclear whether the visual gridworld is a good environment choice for evaluating GVFs. For one, the feature set is not entirely meaningful as the states / observations provided to the agent is the entire grid as opposed to some local region of the agent. Second, while the environment seems simple, the number of iterations needed to learn anything meaningful is on the order of hundreds of thousands.
>
> The gridworld was mainly for illustrative purposes as well as to allow for fast experiments (our main results are on Atari and DeepMind Lab). The Gridworld was used to help the reader understand why depth and action conditionality are important. Although it needed 1 million steps to converge, in practice each run finished in a few minutes which was much faster than training on Atari and DeepMindLab.

---

> > ### Comment · Reviewer_FnXv · 2021-08-31
> > **Thank you for the very detailed rebuttal**
> >
> > Thank you for the very detailed rebuttal, I still have one remaining concern regarding clarity.
> >
> > On the first point, I am still a little confused. The targets are not used in your case because, unlike TD networks [Sutton and Tanner, 2004],  you are not updating the weights. However, you are still updating your predictions $y^i_t$ based on future predictions at other nodes in the neighbourhood $y^j_{t+1}$. In this sense, I understand that this is a temporal difference algorithm. I still don't understand the role of $z$ in your work, is it used at all? Perhaps one way to help guide the reader would be to tie the updates back into the example in Figure 1a and lines 95-108, where you discuss some of the semantics of the question network.

---

> > > ### Author Response · Authors · 2021-08-31
> > > **Response to your follow-up question**
> > >
> > > Thank you for engaging in the discussion.
> > >
> > > To answer your question, $z$ is never used in the implementation of the algorithm. The relation between $z$ and $y$ is analogous to the relation between the true and unknown state-value function $v^{\pi}$ and its parameterized approximation $v_{\theta}$. The role of $z$ is to provide a mathematical definition of the semantics of each prediction. However, the true value of $z$ is never observed in practice. Thus we maintain $y$ as an approximation of $z$ in our answer network (which is shown as the right branch of the agent’s output in Figure 1b) and we *do* update the weights (of the neural network) that compute $y$ via TD to approximate $z$ better (just as in a parameterized $v_{\theta}$).
> > > We acknowledge that the notation is a bit confusing here. We will replace $z$ and $y$ with $z^{\pi}$ and $z_{\theta}$ respectively in the revision for better clarity.

---

> > > > ### Comment · Reviewer_FnXv · 2021-08-31
> > > > **Re:Response to your follow-up question**
> > > >
> > > > Yes, you do update the weights of the answer network which produces the predictions. My comment should have said that the adjacency matrix of the question network is not being updated (not the weights). Of course, this is by design, because the question network is randomly generated. I think my confusion stemmed from $W$ denoting the weights of the answer network in [Sutton & Tanner 2004], but this discussion has helped clear up my confusion. Thank you!

---

### Official Review · Reviewer_p2uV · 2021-07-19

**Rating:** 6
**Confidence:** 3

**Summary:**

Summary. Prior works have used auxiliary tasks (such as pixel control, value prediction) to help with representation learning in RL. These auxiliary tasks can be represented in the form of a question network (or TD network) of a generalized value function (GVF). Rather than handcrafting these question networks (of generalized value function), the paper proposes to randomly generate these question networks and use them to help with representation learning in RL. It shows that these randomly generated question networks can lead to better learning than handcrafted auxiliary tasks. Furthermore, the paper shows that only using these randomly generated question networks still leads to a good representation.

**Limitations And Societal Impact:**

It is adequately addressed in the paper

**Main Review:**

Originality: The use of randomly generated question (or TD) networks to aid with representation learning (in RL) is novel.

Quality: The claims in the paper are backed up by experiments on gridworld, Atari, and deepmind lab domain. It contains proper ablation studies on the gridworld domain which help with the understanding of the method.

Clarity: The paper is written clearly. I wish the authors introduced TD networks in a better way by first giving some high-level motivation and examples of special well-known TD networks (like multi horizon value prediction)

Significance: The paper presents a nice and simple way of learning representations in RL with help of TD networks which outperforms existing methods using hand-crafted auxiliary tasks.


**Time Spent Reviewing:**

1 hr

---

> ### Author Response · Authors · 2021-08-10
> **Author Response to Reviewer p2uV**
>
> We thank the reviewer for the detailed comments. We address the specific comments here.
>
> > I wish the authors introduced TD networks in a better way by first giving some high-level motivation and examples of special well-known TD networks (like multi horizon value prediction)
>
> Regarding high-level motivation, we will add the following to the beginning of Section 3.1 in the revision. *"We would like to investigate auxiliary predictions that: 1) can express rich semantics; 2) can capture long-term predictions; 3) can be learned effectively. A family of GVFs with interdependent TD relationships (a.k.a. TD networks) meets all the criteria."*
>
> Regarding examples of well-known TD networks, we discussed how existing auxiliary predictions can be represented by TD networks in L126-131. We will provide additional more intuitive examples in a revision.

---

> ### Author Response · Authors · 2021-09-01
> **Discussion**
>
> Thank you for your positive initial assessment! We would like to kindly remind you to read our rebuttal. If our rebuttal successfully addresses your concerns we hope that you would take that into account. Thanks!

---

### Decision · Program_Chairs · 2021-09-27

**Decision:**

Accept (Poster)

**Comment:**

The paper presents a surprising result that random deep action-conditional predictions when used as auxiliary tasks yield state representations that produce control performance competitive with state-of-the-art hand-crafted value prediction and pixel control auxiliary tasks in both Atari and DeepMind Lab tasks. All reviewers unanimously vote to accept the paper which I agree with.

In the camera-ready version, I suggest the authors to address the reviewer's comments. It would also be nice to see how the performance gap changes as a function of how of powerful the on-policy learner is. E.g., with A3C / PPO, do we have a similar performance gap? This will help verify whether the same advantage can be gained by improving the policy gradients or not.